# WinoGAViL: Gamified Association Benchmark to Challenge Vision-and-Language Models

**Yonatan Bitton**[†*]  **Nitzan Bitton-Guetta**[‡*]  **Ron Yosef** [†]  **Yuval Elovici**[‡]
**Mohit Bansal**[¶]  **Gabriel Stanovsky**[†]  **Roy Schwartz**[†]

[†]The Hebrew University of Jerusalem   [‡]Ben Gurion University
[¶]University of North Carolina at Chapel Hill
{nitzangu,elovici}@bgu.ac.il; mbansal@cs.unc.edu
{yonatan.bitton,ron.yosef,gabriel.stanovsky,roy.schwartz1}@mail.huji.ac.il

## Abstract

While vision-and-language models perform well on tasks such as visual question answering, they struggle when it comes to basic human commonsense reasoning skills. In this work, we introduce WinoGAViL: an online game of vision-and-language associations (e.g., between *werewolves* and *a full moon*), used as a dynamic evaluation benchmark. Inspired by the popular card game Codenames, a "spymaster" gives a textual cue related to several visual candidates, and another player tries to identify them. Human players are rewarded for creating associations that are challenging for a rival AI model but still solvable by other human players. We use the game to collect 3.5K instances, finding that they are intuitive for humans (>90% Jaccard index) but challenging for state-of-the-art AI models, where the best model (ViLT) achieves a score of 52%, succeeding mostly where the cue is visually salient. Our analysis as well as the feedback we collect from players indicate that the collected associations require diverse reasoning skills, including general knowledge, common sense, abstraction, and more. We release the dataset, the code and the interactive game, allowing future data collection that can be used to develop models with better association abilities.[1]

## 1  Introduction

Humans can intuitively reason about how a cue is associated with an image [1, 2, 3]. For example, in Figure 1, the word *werewolf* may be intuitively associated with images of a puppy and a full moon. These reasoning skills go beyond object detection and similarity and require rich cultural and world knowledge. Cognitive studies suggest that this kind of associative thinking involves connecting distant concepts in the human memory, organized as a network of interconnected ideas [4, 5, 6, 7, 8]. On the other hand, vision-and-language models often fail when faced with tasks that require commonsense reasoning and cultural knowledge [9, 10, 11, 12], motivating the construction of a challenging high quality vision-and-language benchmark.

In this work, we introduce a **G**amified **A**ssociation benchmark to challenge **Vi**sion-and-**L**anguage models (WinoGAViL). Inspired by Winograd Schema Challenge [13], we suggest WinoGAViL as a benchmark for multimodal machine commonsense reasoning and association abilities. Similar to the Codenames game,[2] each instance in WinoGAViL is composed of a textual cue, a number $k$, and a set of candidate images. The task is to select the $k$ images most associated with the cue. We refer to the

---

[*]Equal contribution.
[1]**https://winogavil.github.io/**
[2]https://en.wikipedia.org/wiki/Codenames_(board_game)

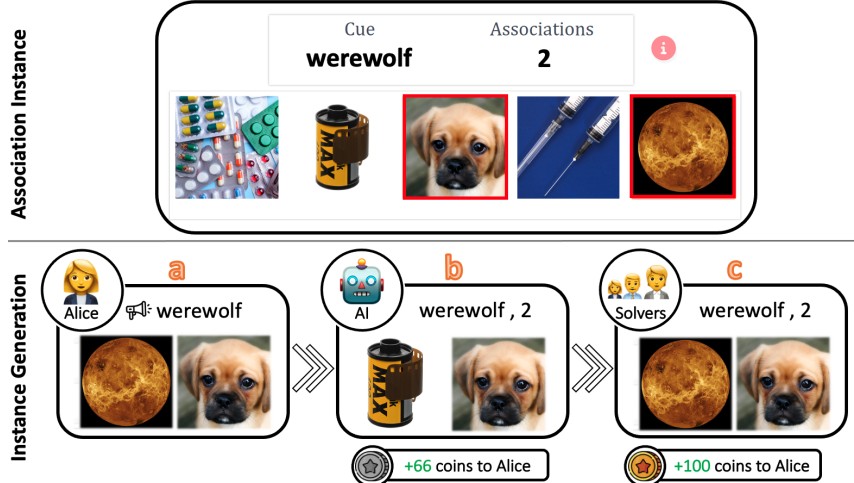

Figure 1: Top: An association instance from the WinoGAViL benchmark. The task is to choose the top $k$ images that suit the cue word. In this example, the top $k$=2 images that suit the cue *werewolf* are surrounded by red bounding boxes. Bottom: Game Setup—a new association instance generation. A spymaster (Alice) composes a new association given a set of images that is challenging for the rival AI model but easy for other human players. (a) Alice generates a cue word for a subset of the images; (b) A rival AI model makes a prediction based on the given cue, and Alice is rewarded inversely to the model performance; (c) Three human solvers also try to solve the task and the spymaster is rewarded according to their performance.

cue and the associated images as an *association instance*. For example, in Figure 1, the pictures of a *puppy* and a *moon* are (arguably) the ones most associated with the cue *werewolf* out of the given candidates.

We develop an online game to collect novel and challenging associations. The game is used to collect data for this work, but more importantly— to serve as a dynamic source for additional data in the future. As exemplified in Figure 1, a "spymaster" first composes a new association cue given a set of images. A rival AI model (CLIP RN50 [14]) then predicts the given association, and the spymaster is rewarded inversely to its performance, motivating the spymaster to make the cue challenging. Lastly, three human players attempt to solve the association task. The spymaster is rewarded according to their performance, motivating the spymaster to compose associations that are solvable by humans and, thus, ideally more natural than examples designed to fool a model. We use crowdworkers to collect 3.5K test instances. See Figure 2 for a collected example.

We evaluate several state-of-the-art models on WinoGAViL data. We find that our game allows the collection of associations that are easy for humans (>90% Jaccard index) and challenging for models (∼52%), even those that are orders

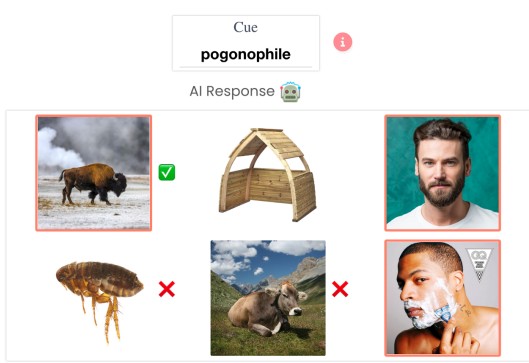

Figure 2: The spymaster screen for an example collected via the WinoGAViL benchmark. The spymaster submitted the cue 'pogonophile' (a lover of beards), and associated it with the three images surrounded by red bounding boxes. Model predictions are marked with V for success and X for failure. In this example the spymaster has managed to partially fool the AI model, while three other humans are able to solve it perfectly.

of magnitude larger than the model used to create the game. Our analysis shows that models succeed mostly where the cue is visually salient. Finally, we compare our collected data with data we collected via an alternative data generation baseline that relies on SWOW [15], a hand-crafted resource of textual associations. Our results show that while the two approaches are relatively easy for humans,

data generated by WinoGAViL is much more challenging to machines, highlighting the value of our gamified data collection framework.

## 2 The WinoGAViL Benchmark

We start by presenting the game as a framework for collecting challenging associations (§2.1). Second, we describe how we crowd-source a test set using the game (§2.2). Finally, we analyze the collected dataset and provide statistics (§2.3).

Throughout this paper we use the Jaccard index, which is the intersection of selected candidates divided by the union of selected candidates.[3] This metric does not reward random guesses highly. The random expected Jaccard index is 38%, 34%, 24%, 17% with 5/6/10/12 candidates respectively. For example, in Figure 1c the Jaccard index ('Human score') of the solvers is 100%, since the intersection of the selections is the same as the union. In Figure 1b the AI model selection is 1/3, so the Jaccard index ('Model score') is 33%: there are three images in the union, and one image in the intersection.

### 2.1 The Game

This section describes the WinoGAViL game environment. Besides collecting the data presented in this paper, the game can also serve as a dynamic source of new data in the future. The game setup is described below in sequential order.

1. **A spymaster creates a challenging association.** A spymaster composes a new *association instance* given a random set of images sampled from the web (see details below). We experiment with sets of 5, 6, 10 or 12 images. The spymaster then submits a single-word cue and selects the subset of associated images. The goal is for the association to be solvable by humans but not by the AI model. For example in Figure 1, the spymaster composes the cue *werewolf* and associates it with the images of the *puppy* and the *moon*.

2. **A rival AI model makes a prediction.** We then feed the association instance to a rival AI model, and report the model score. For example, in Figure 2, the model predicts correctly one candidate (the image of the *bison*), and the total number of candidates involved is 5 :the three images the user selected and the two images falsely predicted by the model. Therefore, the model's Jaccard index is 1/5=20%. The spymaster is rewarded inversely to the model performance, so their "fool-the-AI" score is (100 - 'model score') = 80%.

3. **Three human players validate the created association.** We then give the association to three human validators, who are rewarded according to their Jaccard index for solving the association. Importantly, the spymaster's association "solvable-by-humans" score is determined by the average score of the three solvers. For example, in Figure 1 all players solve the created associations perfectly; therefore, the spymaster's association "solvable-by-humans" score is 100%.

Each player alternates between spymaster and solver roles. Each new association instance created by the spymaster is assigned to three solvers. Once the spymaster creates an association instance, their role changes to a solver responsible for solving other players' associations. This balanced approach ensures that all new associations are automatically validated by three other players.

**Rival AI model.**   We use CLIP [14], with a textual prompt of "A/An [cue]". We intentionally use a small version of CLIP (RN50), so we could evaluate the generated data with larger models. Our experiments (§3) show that this data is indeed challenging for orders-of-magnitude larger models. Future versions of the game will use newer and stronger models, that are likely to further improve the data quality.

**Image extraction.**   We start with a corpus of English concepts obtained from SWOW [15].[4] We collect an image for each concept from Google Images Download. We filter images of written words

---

[3]https://en.wikipedia.org/wiki/Jaccard_index.
[4]We removed words that are potentially offensive using https://pypi.org/project/profanity-filter/.

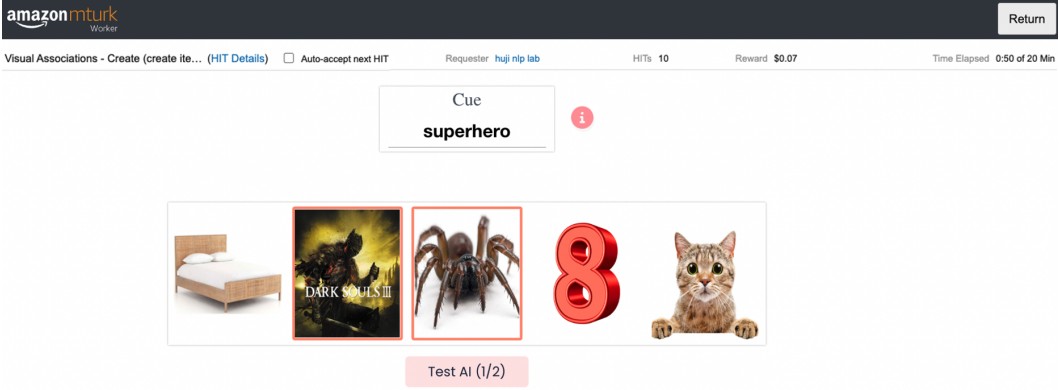

Figure 3: A screenshot from a spymaster screen in Amazon Mechanical Turk.

using an OCR model [16]. We removed images containing the query text using an OCR model (e.g., OCR prediction "brary" for search query "library"). We extract the top image based on google ranking (~2% of the images are filtered). We also manually filter and verify that there are no inappropriate images. The result is a set of 3K images.

**WinoGAViL game properties.** WinoGAViL's main goal is to serve as a dynamic benchmark that remains relevant as the field advances. To achieve this, we publicly release the WinoGAViL web game, allowing dynamic data collection. The players who create associations observe the AI model predictions in real-time. Players switch roles, validating each created association as part of the game. We use rewards to motivate players to create high-quality data according to our metrics. Players are rewarded for both fooling the AI model and making the associations solvable by other humans, preventing the data from becoming unnatural and biased towards only fooling the AI model. The publicly released game includes a player dashboard and a leaderboard. All of these aim to motivate the players to compete with the AI model and with each other, leading to enhanced user engagement and high-quality data.

## 2.2 Human Annotation

We hire Amazon Mechanical Turk workers to play the WinoGAViL game. We develop qualification tests to select high-quality annotators and collect the annotators' demographic information. Spymasters screen example is presented in Figure 3; See Appendix A for more details.[5] We have several options for the total number of candidates: 5, 6, 10 or 12. With more candidates, the task naturally becomes harder. The spymasters are allowed to select between 2-5 images. Full annotation results and statistics are presented in Table 1. The scores of both humans and models is the Jaccard index of between their created associations instances. The annotation task includes three steps, elaborated below.

First, we create new associations by asking three spymasters to create two different cues and associated candidates for a given set of images. The created association should fool the AI model but still be solvable by other humans. To reinforce it, the spymasters receive a bonus payment if their "solvable-by-humans" score is at least 80%, which grows according to their "fool-the-AI" score, see full details of the bonus in Appendix A, Section A.4.1. The first row in Table 1 presents the number of generated associations, and the second row presents the average model score (or 100-"fool-the-AI score"). The low model scores indicate that the spymasters succeeded in creating data that fools the AI model.

Second, we take the associations created via the game and ask three annotators to solve them. We compute an average Jaccard index of the three solvers for each instance. The third row in Table 1 presents the average human score (or the spymaster's "solvable-by-humans" score), indicating that the spymasters were able to create data that is solvable by other humans.

---

[5]We note associations can be subjective and culture-dependent. In Section 3 we show high agreement between our annotators.

Table 2: Some of the skills and observed patterns required to solve WinoGAViL associations. Each association instance may require multiple skills.

| Skill | Observed Pattern | Description | Example | % |
|---|---|---|---|---|
| Non-Visual | Attribute | Cue has attributes of Association
Cue is Association | iguana has green color
miners are dirty | 14% |
| | Use-Of | Cue uses the Association
Association is used in relation to Cue | miner uses tractor
tupperware is used to store food | 9% |
| | General Knowledge | Cue is a name for Association
Association is used in a relation to Cue | ford is a name of a car
Oats improve horses performance | 13% |
| Visual | Activity | Associations perform a Cue in the image | deer & snowman looks like they stare (Figure 6b) | 6% |
| | Analogy | Cue can look like Association, despite being from different concept maps | deer & TV antenna looks like a horn (Figure 6d) | 4% |
| | Visual Similarit | Association is visually similar to the Cue | The sponge shape is similar to a box (Figure 6a) | 20% |

Finally, we select the WinoGAViL test set. To obtain the final test instances, we select associations solved with a mean Jaccard index of at least 80%. The threshold can be lowered to receive more data of lower quality or raised to receive less data of higher quality. Note that in order to reduce the dependence on a specific model, we do not use the model scores in the data selection, i.e., instances that can be solved by the AI model are not automatically excluded, only the solvable-by-humans score is considered in the discarding decision. The last row in Table 1 presents the final number of instances accumulated in the dataset.

The annotators were paid an average of 14 USD per hour for the annotation tasks (including bonuses). The total project annotation budget was 2,000 USD. The annotators received daily feedback on their performances, scores, and the bonuses they won. We denote the data created by the WinoGAViL game by *WinoGAViL dataset*. In §3 we show that this data is easy for humans and challenging for state-of-the-art models.

### 2.3 WinoGAViL Analysis

**Reasoning skills.** We analyze the different skills required to solve the *WinoGAViL dataset*. We randomly sample 320 samples of *WinoGAViL dataset* and manually annotate these skills, observing the patterns required for humans to solve each association. Table 2 presents some of the observed patterns, required skills, and frequencies. Appendix A, Table 8 presents the full table and Figure 6 presents examples of the visual associations. We see that solving *WinoGAViL dataset* requires diverse commonsense skills.

Table 1: WinoGAViL collection statistics. Small differences exist between 5 and 6 candidates, and between 10 and 12 candidates, so we analyze these groups together. Compared to humans, the model struggles with increased number of candidates.

| # Candidates | 5 & 6 | 10 & 12 |
|---|---|---|
| # Generated Associations | 4,482 | 1,500 |
| % Avg. Model Score | 50% | **35%** |
| % Avg. Human Score | 84% | **80%** |
| # ≥80% Avg. Human Score | 2,714 | 854 |

**Players feedback.** We collected qualitative and quantitative feedback from the crowdworkers. Table 3 presents quantitative questions and ratings, showing our game is recommended as an online game, is fun and has an intuitive user interface. We also asked the spymasters open questions about how seeing the AI model prediction and the performance bonus affected them. They mostly responded that these decisions were effective—*"I used the model's guesses to make my associations better. I went after associations that the model frequently got wrong."* and *"bonus keep motivation up when it was hard to come up with connections"*. Full qualitative responses (open text) are presented in Section A.4.2 at Appendix A.

Section A.5 in Appendix A includes additional analysis, for example annotator statistics with demographic information and average performance, and generated cues statistics including richness ratings of the created cues, ratings for abstract and concrete cues, and more.

Table 3: Players feedback collected from the crowdworkers players (scale of 1-5)

| | Rate for the following skills how much you found them required while performing the task | | | | | |
|---|---|---|---|---|---|---|
| Role | Visual Reasoning | General Knowledge | Associative Thinking | Commonsense | Abstraction | Divergent Thinking |
| Spymaster | 4.4 | 3.6 | 4.5 | 3.9 | 4.3 | 4.5 |
| Solver | 4.4 | 4 | 4.7 | 4.3 | 4.1 | 4.1 |

| Role | Interest in play and recommend it as an online game | Level of enjoyment while doing the task | How clear was the UI |
|---|---|---|---|
| Spymaster | 3.8 | 3.7 | 4.7 |
| Solver | 4.1 | 4.4 | 4.9 |

# 3 Experiments

In this section, we provide an extensive evaluation of *WinoGAViL dataset*. First, we show the value of our game, by comparing it to an alternative data generation baseline based on SWOW [17], an existing resource of textual associations. We then evaluate human and models performance on both datasets and provide analysis.

## 3.1 Extracting the SWOW Baseline Dataset

We describe an alternative data generation baseline based on the SWOW ("Small World of Words") dataset.[6] SWOW is an ongoing project where participants are presented with a cue word and asked to respond with the first three words that come to mind. We use a common representation of SWOW as a graph network.[7] We select random distractors that are not associated with the cue in the SWOW graph. We combine the distractors to the association instances from SWOW and create 1,200 multiple-choice instances with 5 or 6 candidates. Each concept's image is obtained from the extracted images (§2.1). Note that SWOW is based on textual associations, which were provided by humans given a cue, making it textual and non-adversarial, whereas WinoGAViL is based on visual and adversarial associations, where humans create a new cue given a set of images. Figure 4 illustrates this difference. As we did in the WinoGAViL game, we validate with human annotation and only keep instances with a mean Jaccard score of at least 80%. Human performance is 85%, so most association instances are retained. The final dataset, denoted *SWOW vision baseline dataset*, is composed of 1,000 instances.

## 3.2 Evaluation Setup

We experiment with state-of-the-art models and compare them to humans on the *WinoGAViL dataset* and the *SWOW vision baseline dataset*. On the *WinoGAViL dataset* we compare cases with 5-6 candidates and cases with 10-12 candidates. We use the Jaccard index as an evaluation metric (§2).

**Humans.** We sample 10% of the test sets and validate it with new annotators who were not involved in any previous annotation tasks. We require three different annotators to solve each instance and report their average Jaccard score as the final human prediction. Annotator agreement is measured two different ways: by comparing the Jaccard index of the annotators' selections with the ground-truth labels, and by comparing the Jaccard index between the three annotators' selections. The standard deviations are 6.3, 7.5, and 5, and the Jaccard index is 80, 81, and 89 for the cases with 10-12 candidates, 5-6 candidates, and SWOW, respectively, indicating high agreement.

**Zero-shot models.** We evaluate several diverse state-of-the-art vision-and-language models. In all cases described below (except CLIP-ViL), the model encodes the text and the image and produces a matching score for each (cue, image) pair. We take the $k$ (number of associations) images with the top scores (For example, the top $k=3$ model predictions in Figure 2).[8]

1. CLIP [14] is pre-trained with a contrastive objective that can be used without directly optimizing for the task. We use four versions of models with different amounts of parameters: RN50, ViT-B/32, ViT-L/14 and RN50x64/14 with 100M, 150M, 430M and 620M parameters respectively (RN50 was used during data collection).

---

[6] licensed under a Creative Commons Attribution-NonCommercial-NoDerivs 3.0 Unported License.
[7] https://smallworldofwords.org/en/project/explore
[8] We ran the zero-shot experiments on a MacBook Pro laptop (CPU) in <6 hours.

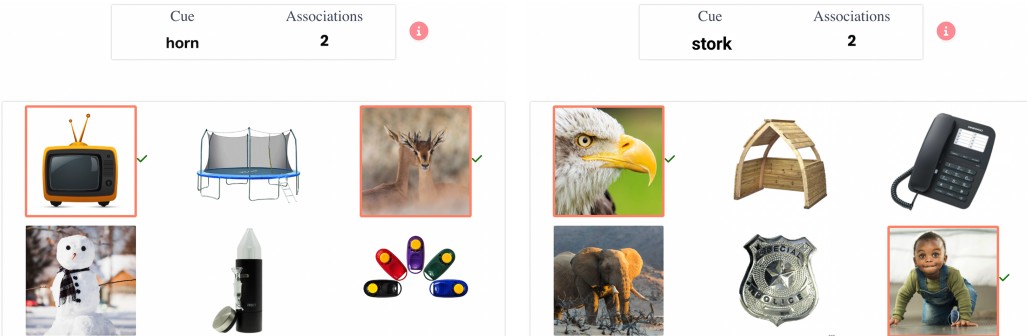

(a) An association from *WinoGAViL dataset*, collected by our interactive game. Spymaster composes a new association given a set of images, aiming to fool a rival AI model. The spymaster has created the cue *horn* and selected the two images surrounded by bounding boxes. This association instance cannot be solved without the specific image information (TVs usually don't have horns). Cues are assigned in a visual and adversarial manner.

(b) An association from *SWOW vision baseline dataset*, which is automatically extracted based on the SWOW dataset. Annotators receive a cue (e.g., *stork*) and provide three associations. We take the textual annotations, add distractors and extract images for each given association. This association could be solved without the visual information (*stork* is correlated with the concepts of *bird* and *baby*). Cues are assigned in a textual and non-adversarial manner.

Figure 4: *WinoGAViL dataset* vs. *SWOW vision baseline dataset* generation process.

2. CLIP-ViL [18], with 290M parameters, is a pre-trained vision-and-language model that uses CLIP as a visual backbone, rather than CNN based visual encoders that are trained on a small set of manually annotated data. We use the image-text matching objective, where a classification head predicts a score indicating whether the candidate image and the cue match each other.

3. ViLT [19], with 111M parameters, incorporates text embeddings into a Vision Transformer (ViT).

4. X-VLM [20], with 216M parameters, is pre-trained with multi-grained vision language alignments and fine-tuned for image-text retrieval (Flickr30 [21]) tasks, achieving state-of-the-art results on several benchmarks.

**Supervised models.** We join a line of benchmarks that introduce a test set, without predefined train splits [10, 22, 23]. We believe that in order to solve associations, a machine must map knowledge to new, unknown cases without extensive training [24]. Nonetheless, for completeness, we also consider fine-tuning models on the associations data. We add a binary classifier on top of the pre-trained embeddings to classify whether a given (cue, image) pair is associative or not. We use CLIP (VIT-B/32) model, concatenate the textual cue embedding to the visual image embedding, followed by a classifier that produces a score in $[0, 1]$, where 1 is labeled 'associative'. We use the Adam optimizer [25] with a learning rate of 0.001, batch size of 128, and train for 7 epochs. Since we do not propose a training/validation/test split, we repeat five experiments with different random seeds where we sample a unified training set of 9,326 (cue,image) pairs for both the candidates cases. We then sample a separate test (10%) and validation (10%) sets with non-overlapping images, and report the average results, comparing the supervised and zero-shot models on the same sampled test sets.[9]

---

[9]Code for reproducing these experiments is available in this link. We ran the supervised experiments with a single NVIDIA RTX2080 GPU, all experiments ran in <24 hours.

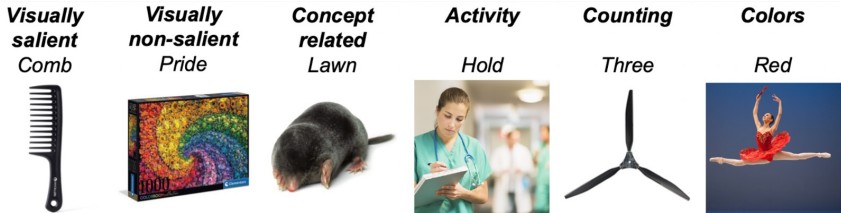

Figure 5: Examples for different association categories and results for each category

### 3.3 Results and Model Analysis

Zero-shot results on *WinoGAViL dataset* and the *SWOW vision baseline dataset* are presented in Table 4. Table 10 (Appendix A) shows full statistics and performance for the different number of candidates and created associations.

**The game allows collection of associations that are easy for humans and challenging for models.** Performance on the data collected via the game is 15–52% with 10-12 candidates, and 47–55% with 5-6 candidates. All models' performances are far below human performance (90% and 92%, see last row). We highlight that although our rival AI model is CLIP with RN50, the created data is still challenging even for models order-of-magnitude larger. We also see a significant performance drop with most models when increasing the number of candidates without hurting human accuracy, indicating that humans are robust to the increased difficulty level while models struggle with it.

Table 4: Zero-shot models performance on the *SWOW vision baseline dataset* and the *WinoGAViL dataset*. Numbers indicates Jaccard score (0–100%). Bold numbers indicate best models performances and lowest human performance. The associations collected via the game are difficult for all models to solve.

| Model | Game | | SWOW |
|---|---|---|---|
| # Candidates | 10 & 12 | 5 & 6 | 5 & 6 |
| CLIP-RN50x64/14 | 38 | 50 | 70 |
| CLIP-VIT-L/14 | 40 | 53 | **74** |
| CLIP-VIT-B/32 | 41 | 53 | 74 |
| CLIP-RN50 | 35 | 50 | 73 |
| CLIP-ViL | 15 | 47 | 66 |
| ViLT | **52** | **55** | 59 |
| X-VLM | 46 | 53 | 68 |
| Humans | **90** | 92 | 95 |

**The game creates more challenging associations compared to the SWOW based method.** The highest model performance on the *SWOW vision baseline dataset* is 74%, and on the *WinoGAViL dataset* is 55%, both with the same number of candidates (5 & 6). CLIP-ViL achieves lower results, especially in the 10 & 12 case. The reason could be that CLIP-ViL uses the ITM pre-training objective (image-text matching), whereas X-VLM and ViLT are fine-tuned for image-text retrieval. CLIP is also pre-trained, but with a different contrastive pre-training objective that may be more useful for this task. The results indicate the value of our game in collecting associations that are much more challenging than the SWOW-based method.

**Training is effective given more distractors.** Fine-tuning results are presented in Table 5. The relatively low performance indicates that models struggle to capture the information required to solve challenging associations from supervised data. Interestingly, we see that training did not change with 5 & 6 candidates, but did improve performance by 7% with 10 & 12 candidates, indicating that the model is only able to exploit supervised data in particularly hard cases, with lower random chance of success.

Table 5: Supervised models performance. Results are mean and standard deviation of the Jaccard index of five experiments, each time sampling different test set. Training is effective given more distractors.

| # Candidates | 10 & 12 | 5 & 6 |
|---|---|---|
| Zero-Shot | 42 ± 3 | 53 ± 2 |
| Supervised | 49 ± 3 | 52 ± 1 |

**Model performance varies between different association types.** We provide a fine-grained model analysis of different association types. We hypothesize that models perform better on association instances that require direct visual detection, as

these models' training objectives are similar to these kind of tasks. We sampled ∼1K cue-image pairs of the instances created via the game with 10-12 candidates for analysis. Three of the authors identified the following six categories: (a) *Visually salient*: the cue is the main visually salient item in the image; (b) *Visually non-salient*: the cue is visual but non-salient, and specific to the particular given image; (c) *Concept related*: the cue is related to the image *concept*, not necessarily to the particular given image; (d) *Activity*: the cue is an activity depicted in the image, e.g., the cue is *jumping*, and the image shows people jumping; (e) *Counting*: the cue is a number or amount of items depicted in the image (e.g., *two* with an image of two people); (f) *Colors*: the cue indicates a color that is depicted in the image (e.g., *white*); Examples are presented in Figure 5. Additional details, annotations guidelines, full examples for each category and screenshots from the annotation task are provided in Appendix A, Section A.7. We define the final category as the annotators' majority vote, that was reached in 98% of the cases, and discarded the other 2%. We evaluate CLIP ViT-B/32 on the association instances and report the accuracy per category which is the proportion of the model success in each given category. Results are presented in Figure 5. We find that model performance is highest in the visually salient and colors category, degrades in concept related, and activities, and is much worse in the visually non-salient and counting categories. The results suggest a lack of common sense reasoning capabilities. We release the annotated data in the project website for future research.

**Performance of textual models is close to vision-and-language models, but still far from human.** Another approach for tackling WinoGAViL is using textual models, when transferring the visual modality to textual modality with image captions, receiving a full-textual dataset. We take OFA [26], a state-of-the-art image captioning model, and extract image captions for each of the image candidates. We use the three leading models for semantic search in Sentence Transformers [27], which are Distilled RoBERTa, [28] and MPNet [29] (two versions, the original model, and a model fine-tuned for semantic search).[10] Results are presented in Table 7. We see

Table 6: Results for different association categories and results for each category. The model (CLIP ViT-B/32) is stronger when the cue is visually salient in the image (a), but weaker in the other cases, especially in visually non-salient associations.

|  | # Items | % Model | % Humans |
| --- | --- | --- | --- |
| Visually salient | 67 | 75 | 98 |
| Visually non-salient | 379 | 36 | 93 |
| Concept related | 426 | 65 | 92 |
| Activity | 24 | 42 | 96 |
| Counting | 25 | 36 | 97 |
| Colors | 14 | 79 | 96 |

that the results are better than chance level, a bit lower than the textual cue and visual candidates' version (ViLT, one line prior to the last), but still far from human performance. These results hint that WinoGAViL cannot be trivially solved by mapping the images to text.

## 4 Related Work

**Associations and Codenames.** Several works have studied the popular Codenames game in the context of natural language processing [30, 31], which is also related to works on semantic relatedness [32, 33, 34, 35]. In the context of associations, a recent work have proposed to use the SWOW resource to evaluate pre-trained word embedding [17], and some works evaluate models with a CNN-based visual components [1, 2]. We expand these ideas to evaluate state-of-the-art vision-and-language pre-trained models.

**Commonsense.** Commonsense reasoning is a topic with increasing interest lately [36]. Many commonsense reasoning tasks have been proposed, both in NLP [37, 38, 39, 40, 41, 42], and Computer Vision [43, 44], including works that require understanding social cues [45, 9]. In the text domain, a number of Winograd Schema Challenge Datasets have been proposed as alternatives for the Turing test [13, 46, 47, 22, 23]. In the vision-and-language domain Thrush et al. [10] have proposed a dataset that tests compositional reasoning in vision-and-language models with the task of matching a caption with its correct image. WinoGAViL also measures vision-and-language reasoning, but focuses on commonsense-based image-cue associations, and primarily serves as a dynamic benchmark as playing the game allows future data collection.

---

[10]https://www.sbert.net/docs/pretrained_models.html

**Human-and-model-in-the-loop.** Models are often used in dataset collection to reduce dataset biases or to create adversarial instances [38, 39, 48, 49, 50], which might limit the created instances to be effected by the used model. For example, in works that create adversarial visual question answering instances [51, 52], human annotators are prompted to fool the model iteratively for each instance, receiving online feedback from the model, and their annotation is allowed to be submitted only after they succeed or after a certain number of trails. In contrast, in our work, the annotators have only one chance to trick the AI model for a given instance. They cannot iteratively 'squeeze' the model to produce an adversarial example. Thus, the generated data is less dependent on the particular AI model since the model is only used to motivate the human player to fool it. In particular, we do not use the models' predictions to choose the test set instances.

Table 7: Results of textual models when using textual image captions for the candidates. ViLT performance (textual cue and visual candidates) performance appear one line prior the the last. Image-to-text might be beneficial, but still far from human performance.

| Model | Game | | SWOW |
|---|---|---|---|
| # Candidates | 10 & 12 | 5 & 6 | 5 & 6 |
| MPNet | 39 | 52 | 72 |
| MPNet QA | 47 | 55 | 75 |
| Distil RoBERTa | 37 | 50 | 65 |
| ViLT (V&L) | 52 | 55 | 59 |
| Humans | 90 | 92 | 95 |

**Gamification.** Gamification was previously used for several purposes, such as data collection [53, 54, 55], education [56, 57], and beat-the-AI tasks for AI model evaluation [58, 59, 60]. Talmor et al. [12] proposed a gamification framework to collect question answering instances. Kiela et al. [61] proposed a dynamic benchmark that supports human-and-model-in-the-loop. We propose a game that serves as a dynamic benchmark of vision-and-language associations, gamifying both human interactions with an AI model and human interactions with other humans.

## 5 Limitations and Conclusions

Despite our efforts to filter inappropriate concepts and images, some players may feel harmed when they are exposed to new generated cues, or when seeing an image that have passed the automatic and manual filtering. Players are able to mark such cases (with a designated 'report' button), leading to immediate removal until further examination. Additionally, players will agree to a consent form when they register. When designing the game, we had several choices to make, including the bonus reward and the AI model interaction. Future work will thoroughly explore the impact of these choices.

We introduced an online game to collect challenging associations. We demonstrated its effectiveness by collecting a dataset that it is easy for humans and challenging for state-of-the-art models. We provided an extensive evaluation of the game and collected dataset. We hope the WinoGAViL benchmark will drive the development of models with better commonsense and association abilities.

## Acknowledgements

We would like to thank Moran Mizrahi for a feedback regarding the players survey. We would also like to thank Jaemin Cho, Tom Hope, Yonatan Belinkov, Inbal Magar and Aviv Shamsian. This work was supported in part by the Center for Interdisciplinary Data Science Research at the Hebrew University of Jerusalem, and a research grant no. 2088 from the Israeli Ministry of Science and Technology.

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
