# OpenReview forum: "WinoGAViL: Gamified Association Benchmark to Challenge Vision-and-Language Models"
_NeurIPS.cc/2022/Track/Datasets_and_Benchmarks — NeurIPS 2022 Datasets and Benchmarks _

### Official Review · Reviewer_gLGp · 2022-07-27
**An interesting benchmark for testing the commonsense reasoning of vision-and-language models**

**Rating:** 7
**Confidence:** 4
**Clarity:** Yes

**Strengths:**

(1) The WinoGAViL is an interesting benchmark for multimodal commonsense reasoning, and it can be dynamically updated with the proposed gamification framework.

(2) The experiments well demonstrated the gap between human and machine performances, implying the challenges of the designed tasks for current AI methods.

(3) The analysis of the tasks in the WinoGAViL dataset reveals that a diverse set of reasoning skills are required to achieve high performance.

(4) An alternative approach to collecting the data using the SWOW dataset is presented in this work to show that the gamification framework creates more challenging associations.


**Weaknesses:**

(1) It is unclear why not use the model scores in the data selection. My intuition is that removing data that can be easily solved by the AI model (i.e., the CLIP-RN50) will make the benchmark more challenging to current vision-and-language models. Is that true? If not, I wonder how to explain this phenomenon and whether there is any experiment to justify it.

(2) In experiments, it would be more interesting to show how various methods behave differently across tasks that require different reasoning skills. It is an extension of the existing experiments of comparing “non-visual” and “visual” settings to more fine-grained observed patterns.

(3) What are the chance performances in Table 4? I am curious how the methods compare with the random guess.


**Additional Feedback:**

See the above comments

**Correctness:**

The benchmark is constructed in a sound way and the evaluation methods and experiment design well support the claims.

**Documentation:**

Yes

**Relation To Prior Work:**

Yes

**Summary And Contributions:**

This paper proposed a new benchmark called WinoGAViL to evaluate the commonsense reasoning ability of the state-of-the-art vision-and-language models. In WinoGAViL, each task is composed of a textual cue, a number $k$ and a set of candidate images, and its goal is to select the $k$ images most associated with the textual cue. To collect novel and challenging associations, this work proposed a web gamification framework where a spymaster composes a new association cue that can be easily solved by other human players but can easily fool the AI model (such as CLIP-RN50). Experiments showed that current vision-and-language largely fall behind human performance.

---

> ### Author Response · Authors · 2022-08-08
> **Reviewer gLGp response**
>
> We are happy that the reviewer found our benchmark interesting for multimodal commonsense reasoning, that the experiments well demonstrate the gap between humans and machines, and imply the challenges for current AI methods.
>
> **Why not to use the model scores in the data selection**:
> This is a great point that other reviewers asked as well. Please refer to General Response #1: Not discarding associations that are solvable by the AI).
>
> **Analyzing how various methods behave differently across tasks that require different reasoning skills**:
> We focused on a better model evaluation of different association types, with a defined annotation task, more data (~1K samples), three annotators, and also publish this annotated data. Please see General Response #2: Model analysis on different association types.
>
> **Chance performances**:
> Please see Footnote 4 (in both the old and revised version): The random expected Jaccard is 38% / 34% with 5 / 6 candidates, and 24% / 17% with 10 / 12 candidates. It is computed with binomial distribution analysis. For example, with 5 candidates, given 2 selected associations, the probability to be correct in 0 guesses is 0.3: to fail in the 1st guess is 3/5, multiplied by the probability to fail in the 2nd guess - 2/4. We did the same computation with all other options.

---

> > ### Comment · Reviewer_gLGp · 2022-08-29
> > **Reply to Author Rebuttal**
> >
> > Thank the authors for providing a detailed response, which addresses my major concerns. I keep my initial rating and recommend an acceptance.

---

### Official Review · Reviewer_o1Pk · 2022-07-27

**Rating:** 7
**Confidence:** 4
**Clarity:** The paper is very well written, and e…

**Strengths:**

Strengths:
1. The idea is a very clever one, and the codenames-style data collection game is well formulated to collect difficult vision-language associations.
2. The framework and game are made public, so that people can continually add associations.
3. The evaluation of the collected associations is well done, and the results are convincing that these associations are harder to solve for existing VL models but highly solvable by humans (but still less than the SWOW Baseline)
4. The paper is very well written and structured. It was honestly a pleasure to read.


**Weaknesses:**

1. I find it odd that the authors collect only one image per concept from Google Images (L100). Collecting only one image per concept results in cues for concepts that are too specific to the single image collected for it. This means that the cues. corresponding to the various associations that a concept occurs in, largely contain information about the specific visual context of the concept's image, and so the concept's evaluation overfits to the visual properties of that image. While collecting a larger set of images (say, 5) per concept would also result in cues that are specific to the images that in their instances, since the same image is not used for each instance the overall evaluation for that concept's associations does not overfit to a single image.

2. The various visual skills in Table 2 only make it more confusing exactly what we are evaluating with the collected associations. Do snowmen stare? Is this a general association that we want the model to know? It seems like the associations being collected are too image-specific (related to point 1 above). Even if we only want to evaluate the associations that the model can express conditioned on a visual context (i.e. not a general concept-level association like "snowmen are white" but an image-specific one like snowmen staring in that particular image), the overall evaluation still only evaluates associations specific to that one image of the snowmen - how do I know if the model will express snowman-related associations for a different image of the snowman?

3. Further, it's not clear what visual associations for all concepts derived from SWOW will look like. For instance, what do the images of "green" and "dirty" (concepts from the first row of Table 2) look like? How does that affect the particular associations that we collect for them?

4. The choice to not discard associations that are solvable by the AI (L148) is confusing - the whole point of the framework is to collect associations that are difficult for models but solvable by humans, so why keep samples that do not meet the first criteria? The authors could average model performances from a few different AI models if they are concerned about model dependence of their collected data.

5. It's not clear in Section 3.1 how the SWOW Baseline dataset is different from the WinoGAVIL one - specifically, what are properties of how the Baseline is collected that WinoGAVIL does better? (the explanation in L253-258 would go well in S3.1, because it's not clear how these datasets differ till we compare the results on them) Some visual examples that compare associations from these two datasets would be nice.

6. In L274-276, when filtering visually salient cues, I don't understand why the authors do not retain only those cases where both the annotators agree - 88% is still a sizable number, and gives us more confidence about the results on this subset — in the current setup, the results appear to be muddled by those examples where annotators disagreed on the visual salience of the cue.

**Additional Feedback:**

-

**Correctness:**

I have questions about certain design decisions when creating the benchmark, which I have outlined above.

**Documentation:**

Yes

**Ethics:**

None that I see

**Relation To Prior Work:**

Yes

**Summary And Contributions:**

The authors introduce WinoGAViL, a game which acts as a framework to collect language-vision associations that are difficult for existing vision-language models to solve but solvable by humans. The authors design WinoGAViL in the style of Codenames, where the "spymaster" is provided a set of images, and has to come up with a language cue that is related to a certain subset of the images. The authors initially collect a set of 3.5K association instances, but more instances can be flexibly added to this. The authors analyze the collected associations and the challenges involved in solving them, and analyze how existing vision-language models perform on this dataset.

This paper has one of the most novel and intriguing ideas I have seen in vision-language data collection, but after reading the paper and playing around with the demo myself, I have a few doubts that make me skeptical of this work until they are addressed. The strengths of this paper are very strong ones, but unfortunately, the weaknesses in certain methods are also glaring. If the questions listed are adequately answered, I would be willing to change to Clear Accept.

---

> ### Author Response · Authors · 2022-08-08
> **Reviewer o1Pk response**
>
> We thank the reviewer for acknowledging that "this paper has one of the most novel and intriguing ideas I have seen in vision-language data collection". We will do our best to address your concerns.
>
> **One image per concept from Google Images**:
> Thank you for the interesting and thoughtful suggestion. We agree that collecting more images per concept will further diversify the data. We will consider doing so in the next version of the dataset. Below we discuss the proposal in detail.
>
> **Image specificity**:
> To your first question, yes, the associations we collect are indeed image-specific. We see it as a feature rather than a bug. The model must analyze each example and determine whether the cue is associated with it, including the context in which it appears. For instance, we wouldn't expect an image of a snowman shown from behind to be associated with the "stare" cue. Your feedback motivates us to make a more fine-grained partition of the generated associations. Please see General Response #2: Model analysis on different association types (lines 275-295) in the revised version. Additional categories were defined, such as an association based on a concept and not on an image (Figure 14 in the revised version), and an association based on a particular image (Figure 13).
>
> Although the associations are image-specific, we argue that the risk of overfitting the visual properties of that image is small. First, please note that the concepts used to collect images were only used for generating the initial seed of images and are never used in the game. In this sense, there is no immediate risk of associating snowmen with staring, as the concept of “snowman” is not part of the game. Second, our main evaluation setup is zero-shot, so the model is never trained on the images; therefore, overfitting is not an issue in this setup. Third, even in the (secondary) supervised setting, the train/dev/test images are disjoint, so the model is never tested on the same cue with the same image.  Fourth, we refer to some statistics we provided in the paper and extract additional statistics to address your concern:
> 1. Lines 774-777 in the previous version (805-807 in the revised version): “we measure how often different annotators compose the same cues for the same group of images. Since we asked three different annotators to provide two different cues for each group of images, we have six annotations for each image group. We find that almost always (98%) they combine different cues”. This point indicates that humans can usually come up with new and distinct cues for a given set of candidate images at the association instance level.
> 2. We note that the average number of cues per image is 4.34; for example, the image of the concept “tobacco” is linked to the following four cues: expired, brand, unhealthy, and snuff. This point indicates that humans created different variant cues in the single image level, and there is no one-to-one mapping between the cues and the image.
> 3. In 97% of the cases, the (cue, image) pair is unique to its association and doesn't appear in other association instances, indicating a diversified data collection.
> 4. Our last experiment (L294, Table 6) shows that trying to infer the visual properties with image captions still leads to low performance. This may be explained by the fact that the image collection produces images that are difficult to link back to the concepts from which they were sampled.
>
> **What visual associations for all concepts derived from SWOW will look like, and the difference from WinoGAViL**:
> Thank you for emphasizing this point. We started from SWOW concepts and used Google Image Search to extract the images. For example, SWOW has the cue “iguana” associated with “green.” We extracted an image for the “green” concept (which is simply an image with a green background) and the same for the concept “dirty” (which shows an image of dirty hands if you look for it in Google images). In Table 6, we show that humans solve the SWOW baseline dataset with high performance (95%). The main difference is that SWOW started from textual associations. Thus SWOW uses only textual and non-adversarial associations, whereas WinoGAViL uses visual and adversarial associations, where humans create new cues given a set of images. We included this explanation and provided a visual example illustrating the differences in the revised version; please see lines 189-192 and Figure 3 in the revised version.
>
> **Not to discard associations that are solvable by the AI**:
> This is a great point that other reviewers asked as well. Please refer to General Response #1: Not discarding associations that are solvable by the AI.
>
> **Retaining cases where both the annotators agree**:
> You are correct; we retained only the cases agreed by the annotators. We clarified it in the revised version; please see line 290 in the revised version.

---

> > ### Comment · Reviewer_o1Pk · 2022-08-16
> > **Response to Rebuttal**
> >
> > The reviewers have satisfactorily answered my questions about image specificity and SWOW. The performance of the model on fine-grained association types is a nice analysis.
> >
> > However, I still disagree about retaining associations that the model can solve - I understand the concerns about making the associations model-specific, but I think discarding associations that a set of different models (say, 3) could all solve would be a reasonable compromise. However, the authors point out that the associations are provided with the fool-the-AI score and researchers can filter based on these scores if they choose to, so I am a little less concerned than I initially was.
> >
> > Having played with the WinoGAVIL game on their website myself, I am still concerned that there is only one image per concept. I am not going to get hung up on this point, but I would strongly encourage the authors to include more images per concepts in the next version of the dataset.
> >
> > I have consequently increased my score to 7

---

### Official Review · Reviewer_m9HE · 2022-07-27
**A very interesting paper for evaluating vision-and-language models**

**Rating:** 7
**Confidence:** 3
**Correctness:** I do not find any incorrectness in th…
**Clarity:** This paper is well-written and easy t…

**Strengths:**

1. This paper proposes WinoGAViL, an online game to collect vision and language associations, used as a dynamic benchmark to evaluate state-of-the-art models. To me, the proposed approach is novel and has a great contribution.

2. This paper is well-written and well-motivated and has insightful discussions.

3. This paper also releases the dataset, the code and the interactive game, aiming to allow future data collection that can be used to develop models with better association abilities.

**Weaknesses:**

I do not find any weaknesses in this paper.

**Additional Feedback:**

No.

**Documentation:**

Yes.

**Ethics:**

This paper has no ethics concerns.

**Relation To Prior Work:**

Yes.

**Summary And Contributions:**

This paper proposes an online game WinoGAViL to collect vision and language associations, (e.g., werewolves to a full moon), used as a dynamic benchmark to evaluate state-of-the-art models. Specifically, the paper uses the game to collect 3.5K instances, finding that they are  intuitive for humans (>90% Jaccard index) but challenging for state-of-the-art AI models, where the best model (ViLT) achieves a score of 52%, succeeding mostly where the cue is visually salient, which is interesting and insightful.  Moreover, those experimental results in this paper also indicate that the collected associations require diverse reasoning skills,  including general knowledge, common sense, abstraction, and more. Overall, this paper is well-motivated and well-written. I believe it can promote the development of visual-language models.

---

> ### Author Response · Authors · 2022-08-08
> **Reviewer m9HE response**
>
> We are happy that the reviewer found our paper well motivated and well written, easy to follow, and acknowledging it can “promote the development of visual-language models”.

---

### Official Review · Reviewer_D9dA · 2022-07-28
**A challenging V&L benchmark inspired by Codenames board game**

**Rating:** 9
**Confidence:** 4
**Clarity:** The paper is very well-written. Easy …

**Strengths:**

The idea is quite novel and interesting. I am certain that this benchmark will be a beneficial resource for future researchers. NLP community is interested in problems that require commonsense reasoning and/or external knowledge recently, and this benchmark falls into this category. The problem is easy for humans and challenging for machines (which should be the case for all created datasets). The paper includes many different categorical analyses (e.g. Table 2, Figure 4). Overall, the paper is well-written.

**Weaknesses:**

I can't find any major weakness, but I think the paper has minor issues which can be improved (see the Correctness section).

**Additional Feedback:**

N/A.

**Correctness:**

I think there are several minor issues,
- For the observation of _Models struggle with associations that are not visually salient_: Is there any further analysis on this result? 60% model performance on visually salient examples is still low.
- For the observation of _Training is effective when the task is difficult_: How is the 10 & 12 task more difficult than 5 & 6? Since the humans achieve similar performance on both tasks, could we really say this? This statement contradicts with the _easy for humans, challenging for models_ claim. The models might be benefitting more from negative associations in finetuning (just a possibility), and those tasks contain more negative examples (in terms of percentage). Do you have an analysis of which examples / cases are improved with finetuning?
- For the observation of _Solving data collected with WinoGAVIL with textual models is not beneficial_: This makes me confused because actually (pseudo) text-only baselines perform similar to vision-language baselines.
- Could you explain why does CLIP-ViL perform worse with comparison to the rest of the models? (especially on the 10 & 12 task)
- A suggestion: could you also perform categorical evaluation on the skills and observed patterns (Table 2, great analysis by the way) required to solve the task? In this way, people can better compare the models.

**Documentation:**

Yes, see the supplementary material.

**Ethics:**

There is no need for further discussion or review.

**Relation To Prior Work:**

I think the related work section is adequate.

**Summary And Contributions:**

This paper introduces a new vision-and-language (V&L) benchmark based on gamification. The gamification framework is very similar to Codenames (a board game), and in this benchmark the aim is to associate _k_ images from a set of candidates for given input concept. First, the _spymaster_ selects k images from a set of candidates, and the solvers try to predict selected images. To do so, a game is designed for AMTs, and a dataset is collected from the logs of this game. Finally, a couple of known V&L models are evaluated on this dataset.

---

> ### Author Response · Authors · 2022-08-08
> **Reviewer D9dA response**
>
> We are pleased that the reviewer found our idea novel and interesting and acknowledged that our benchmark “will be a beneficial resource for future researchers”. We appreciate highlighting the minor issues, which we discuss below.
>
> **An analysis for the 60% model performance on visually salient examples**:
> Your suggestion motivated us to make a further analysis which we think is more beneficial now. Thank you. Please see General Response #2: Model analysis on different association types.
>
> **Task difference between the 10&12 and 5&6**:
> This is a good point. We argue that the task with 10 & 12 is more difficult because the score of a random guess is lower when you have more distractors. Moreover, human performance also decreases (though not by much) when moving from 5&6 to 10&12 (Table 4). We do agree that at the same time, models might be getting more training signals when more distractors are available.
>
>
> **Textual models perform similar to vision-and-language baselines**:
> Thank you for the comment; this was indeed a bit confusing. A valid strategy to solve WinoGAViL is to use textual models over captions produced by image captions. We included this experiment for transparency, and it could indicate a bias and artifacts if the results were different, for example, if textual models would reach much better results compared to V&L models, and close to human performance. The results indicate that textual models achieve performance close with V&L models but still far from human performance. Following your feedback, we changed the title in the revised version to “Performance of textual models is close to vision-and-language models, but still far from human”, line 296 in the revised version.
>
> **Why CLIP-ViL performs worse with comparison to the rest of the models**:
> That’s a good point. We hypothesize this is because CLIP-ViL uses the ITM objective (image-text matching), a pre-training objective in which the model receives a single (image-text) pair and determines whether they fit. However, X-VLM and ViLT are fine-tuned for the image-text retrieval task, thus better suited for this task. CLIP is also pre-trained but with a different contrastive pre-training to predict the correct pairing out of N examples, which may be more beneficial for the downstream task. We added it to the revised version; please see lines 259-262.
>
> **Categorical evaluation on the skills and observed patterns**:
> We focused on a better model evaluation of different association types, with a defined annotation task, more data (~1K samples), three annotators, and also publish this annotated data. Please see General Response #2: Model analysis on different association types.

---

### Official Review · Reviewer_5C9t · 2022-07-31
**Interesting trial for better commonsense and association abilities**

**Rating:** 6
**Confidence:** 5

**Strengths:**

- This benchmark is interesting since it highlights the gap of ability for commonsense reasoning between humans and AIs.
- The authors provide a dataset and the online game, allowing researchers to collect further data.
- Experimental results show that the benchmark is challenging for AIs while humans can solve.

**Weaknesses:**

- The game, though inspired by Codename, is a different one. The reviewer is a little concerned that the incentive for spymasters to look for a unified representation for more images may not appear. For example, if the spymaster comes up with a word for two images and the AI misses them all, the Jaccard index is, of course, 0%. If the spymaster thinks of a word for three images and the AI misses them all, the Jaccard index is still 0%. In the original Codename, the number of cards guessed by other than the spymaster is directly scored, so the spymaster thinks of abstract expressions suitable for more cards. However, in the game in this paper, no such incentive is created through the Jaccard index. In order to address human commonsense reasoning, incentives to assign words to more images would have been necessary.
- Although the title includes Vision-and-Language, it is slightly overclaimed since the associated representation seems to be a single word. (If the authors mean that the dataset includes not only words but also sentences as a cue, the reviewer would like them to point this out.) Cues did not have to be limited to words only, as in Codename. In the sense of association and commonsense reasoning, spymasters may generate a more flexible representation for a subset of candidates. Some relations between cue length and performance of AIs and humans would also be interesting.

**Additional Feedback:**

When introducing technical words such as the Turing test and Jaccard index, the authors should avoid referring to Wikipedia since correctness is not guaranteed.

**Clarity:**

There are reasonable descriptions of how the data set was collected and processed and experimental methods.

**Correctness:**

The reviewer thinks this paper has sufficient correctness.


**Documentation:**

Documentation exists for the dataset in the Supplemental Material and on the website.

**Ethics:**

The authors point out a possible concern that players may feel harmed during the game and provide a solution.

**Relation To Prior Work:**

Adequate citations and comparisons to related works have been made.


**Summary And Contributions:**

This paper provides a novel dataset and its code to increase the number of samples. The dataset is named Gamified Association benchmark to challenge Vision-and-Language models (WinoGAViL), inspired by Winograd Schema Challenge [9]. The dataset is collected via an online game resembling the well-known Codename game. In the game, given image candidates, a spymaster associates a cue word for a subset of them. Spymaster's goal is to come up with a set of images and their cue word that AI cannot guess and that humans can guess correctly.

---

> ### Author Response · Authors · 2022-08-08
> **Reviewer 5C9t response**
>
> We are happy the reviewer found our work novel and recognizes that it “highlights the gap of ability for commonsense reasoning between humans and AIs”. We appreciate both suggestions on further improving our game and will consider them in future iterations. We discuss both suggestions below.
>
> **Incentivizing the spymaster to give a large number of cues**:
> We agree. Indeed our game is a bit different from codenames in this sense. As you can see in the statistics provided Appendix 6 Table 9, the dataset does contain significantly more cases with 2 selected images for an association instance compared to 3 or 4 selected images. Followup iterations could consider adding incentives to create association instances with more selected images (e.g., multiplying the score by the number of assigned images).
>
> **Multi-word cues**:
> This is an excellent suggestion that would definitely make the game more interesting! Our design focused on preventing spymasters from simply describing images with multi-word cues, each uniquely describing a different image. For example: using the cue “puppy moon” to describe the two images in Figure 1. Allowing spymasters to write  non-trivial expressions might make the game more interesting, but it would be more challenging to verify that these constitute “legal” cues, and therefore we leave it to future work.

---

> > ### Comment · Reviewer_5C9t · 2022-08-24
> > **Maintained the initial score.**
> >
> > The response from the authors is sufficient.
> >
> > While the authors agree on the weaknesses, they responded that they will address them in future work. The reviewer has already gave the weak accept score, so would like to maintain it.

---

### Author Response · Authors · 2022-08-08
**General response to reviewers**

We appreciate that all reviewers found merit in our work and that WinoGAViL provides a novel method for collecting data and is a valuable and challenging resource for developing vision-and-language models. Please see below how we intend to address the reviewers' comments.

**General Response #1**: Not discarding associations that are solvable by the AI (Reviewers **o1Pk** and **gLGp**).
As for our choice to ignore the model predictions in the data selection, that’s a great point. We would first like to emphasize that the published dataset includes the fool-the-AI score (100 minus the model score) so we can sample a split that considers this score at any point. We had extensive discussions about using this score for data selection, and we decided not to use this score in the released version for the following reasons. We would be happy to hear your input and continue the discussion here. First, as explained in the related work section (lines 336-343 in the revised version; lines 314-321 in the previous version), we wanted the data to be less dependent on the particular AI model used in the game. Based on a sample with low CLIP RN50 scores, it is reasonable to assume that CLIP RN50x64/14 will also produce low results, which may be a consequence of using a similar model for generation and evaluation - which we wanted to avoid. Even if we average the model performances for different AI models, there will still be a model dependence as we see it. Furthermore, the dataset seems challenging for the state-of-the-art models we evaluated. Given more powerful models in the future, we can carefully sample splits that are challenging for a specific family of models and publish them formally.

**General Response #2**: Model analysis on different association types (Reviewers **o1Pk**, **gLGp**, **D9dA**).
As for analyzing cues specificity to images (**o1Pk**), the extension of the existing experiments of comparing “non-visual” and “visual” settings to more fine-grained observed patterns (**gLGp**), and the comment by **D9dA** about cues specificity to the images. These reviews motivated us to expand our previous model analysis of visually salient and non salient cues. We defined an annotation task with multiple fine-grained patterns, more data, and three annotators. Our final category is determined by the majority votes of the annotators, and we publish this data along with annotated guidelines for future research. Based on the results, we can better understand which types of associations the model understands and which types it does not. Please see lines 275-293 in the revised version; additional details in the Appendix 7, lines 814-824, including figures 12-19.

---

### Meta-Review · Area_Chair_NitU · 2022-09-10

**Recommendation:** Accept
**Confidence:** 4

**Metareview:**

The paper presents a new vision-and-language benchmark that associates images and text, collected using a crowdsourced game similar to Codemaster. All reviewers were positive on the paper. They praised the cleverness and novelty of the idea, the clarity of the writing, that the benchmark and game are publicly available, and the quality of the experimental evaluation. Reviewer o1Pk initially raised some issues, but was largely convinced by the authors’ response. Overall the AC agrees that the paper is interesting and well-written, and will be a welcome addition to the NeurIPS Datasets and Benchmarks Track this year.

---

### Decision · Program_Chairs · 2022-09-16

Accept